# DeCodec: Rethinking Audio Codecs as Universal Disentangled Representation Learners

Demo Page: `https://luo404.github.io/DeCodecV2/`

## Abstract

Universal audio codecs learn entangled representations across audio types, whereas some specific codecs offer decoupled representations but are limited to speech. Real-world audio, however, often contains mixed speech and background sounds, and downstream tasks require selective access to these components. Therefore, we rethink the audio codec as **an universal disentangled representation learner** to enable controllable feature selection across different audio tasks. To this end, we introduce **DeCodec**, a novel neural codec that learns to decouple audio representations into orthogonal subspaces dedicated to speech and background sound, and within speech, representations are further decomposed into semantic and paralinguistic components. This hierarchical disentanglement allows flexible feature selection, making DeCodec an universal front-end for multiple audio applications. Technically, built upon a codec framework, DeCodec incorporates two key innovations: a **subspace orthogonal projection module** that factorizes the input into two decoupled orthogonal subspaces, and a **representation swap training procedure** that ensures these two subspaces are correlate to the speech and background sound, respectively. These allows parallel RVQs to quantize speech and background sound components independently. Furthermore, we employ semantic guidance to the speech RVQ to achieve semantic and paralinguistic decomposition. Experimental results show that DeCodec maintains advanced signal reconstruction while enabling new capabilities: superior speech enhancement and effective one-shot voice conversion on noisy speech via representation recombination, improved ASR robustness through clean semantic representations, and controllable background sound preservation/suppression in TTS.

## 1 Introduction

In real-world scenarios, audio often consists of both speech and background sounds (BGS). Speech conveys semantic content and speaker characteristics (Rubenstein et al., 2023), while background sounds provide environmental cues that enhance realism (Wang et al., 2023b). Different audio tasks prioritize these elements differently: speech enhancement (SE) and speech recognition (ASR) only focus on speech while suppressing background sound (Wang, 2025), whereas text-to-speech (TTS), voice conversion (VC), and virtual sound construction treat background sound as valuable for immersion (Yao et al., 2023). Thus, decoupling speech and background sound enables controllable information selection for diverse audio tasks, which is crucial step toward universal audio processing.

Currently, speech-background sound decoupling primarily relies on speech separation (SS) (Wang & Chen, 2018), which aim to obtain seperated signals in the time or time-frequency domain. In traditional cascaded pipelines, SS serves as an independent front-end to extract target signals (Wang et al., 2020; Weninger et al., 2015), after which downstream tasks perform secondary feature extraction (Natarajan et al., 2025; Yang & Chang, 2023).

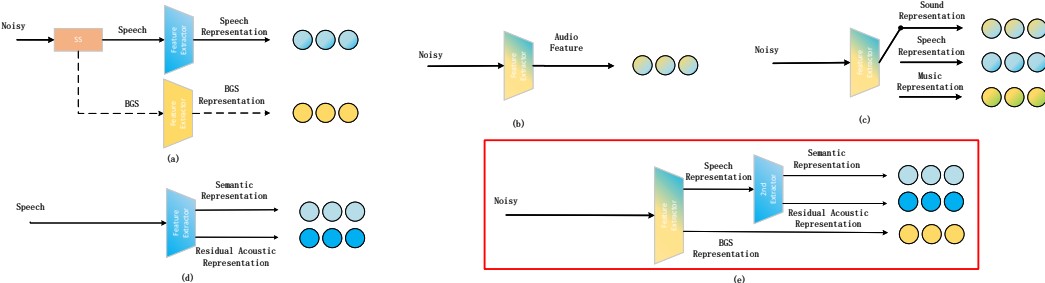

Figure 1: Illustration of different feature extraction methods: (a) cascading pipline, (b) acoustic audio coding, (c) audio classification coding, (d) speech decomposition coding, and (e) the proposed method.

This pipeline is illustrated in Figure 1 (a). However, this pipeline has three key limitations: 1) Back-end performance is highly dependent on front-end separation quality, leading to error propagation; 2) Front-end separation causes target signal distortion, requiring back-end fine-tuning on noisy data which increases training complexity; 3) In multi-task scenarios, deploying dedicated feature extractors per task substantially increases computational cost.

Inspired by human auditory processing: Different regions of the secondary auditory cortex (A2) separately process speech and background sound (Mesgarani et al., 2014; Nourski & Brugge, 2011), allowing higher-order regulatory regions dynamically select task-relevant features based on cognitive demands. This perceptual mechanism motivates us to rethink audio codecs as universal disentangled representation learners that explicitly decouple speech and background sound in the representation domain to enable task-aware control over these features. This proposal offers three core advantages: 1) Preservation of original signal integrity without front-end separation distortion; 2) A unified representation domain enabling parallel multi-task processing; 3) Computational efficiency via feature selection rather than differential extraction.

However, existing neural audio codecs have not yet achieved decoupling of speech and background sound in the representation domain. In the field of universal audio coding, mainstream acoustic codecs such as Encodec (Défossez et al., 2022) and DAC (Kumar et al., 2024) can maintain high audio reconstruction fidelity, but their design is entirely based on signal-level discrete coding and does not explicitly distinguish between speech and background sound, as shown in Figure 1 (b). Lately, UniCodec (Jiang et al., 2025) innovatively coding different types of audio *i.e.* speech, music and sound through a partitioned domain-adaptive codebook, but it classifies noisy speech as 'sound' type and encodes the overall information in a single code, as shown in Figure 1 (c). Therefore, it can only achieve rough audio type classification rather than genuine representation decoupling, and thus cannot actually enable the selection of audio components to different processing tasks.

Compared to universal audio codecs, research on speech representation involves some work on decomposing components. FACodec (Ju et al., 2024) decompose speech into content, pitch, timbre, and acoustic residual. However, there is significant information leakage between these components. Differently, SpeechTokenizer (Zhang et al., 2023), Mimicodec (Défossez et al., 2024), and DualCodec (Li et al., 2025) simply disentangles speech information into semantic and residual acoustic aspects, as shown in Figure 1 (d). However, the performance of these methods in real noisy environments remains limited. Synergistic optimization with background sound decoupling mechanisms offers the potential to enhance their performance in real applications.

Follwing the concept of universal disentangled representations, this study proposes DeCodec, a novel neural codec that simultaneously decouples speech and background sound in the representation domain and collaboratively decouples semantic and residual paralinguistic information within speech, as shown in Figure 1 (e), enabling controllable representation selection for diverse audio tasks. Corresponding to auditory perception mechanisms, to simulate the regional processing of speech and background sound by the A2, we proposes

a subspace orthogonal projection (SOP) module, projecting the primary audio embeddings into two orthogonal subspaces to achieve decoupled representations. Inspired by neural developmental feedback mechanisms (Zhang, 2019), a representation swap training (RST) procedure is proposed to motivate the above two orthogonal subspaces to correspond to the speech and background sound subspaces, respectively. Additionally, during the speech representation quantizing, semantic guidance (SG) is introduced to further guide hierarchical quantizing of semantic and residual paralinguistic information. The above information can ultimately be decoded into time-domain signal via the decoder. The key contributions of this paper are summarized as follows:

- By proposing SOP module and combining it with the proposed RST procedure, the DeCodec achieved explicit decoupling representation of speech and background sound in the feature domain for the first time, realizing an universal disentanglement codec.
- By adopting SG technology and employing a collaborative optimization strategy with speech-background sound decoupling, the DeCodec have enhanced the robustness of semantic and residual paralinguistic representations against background sound interference.

Experimental results show that our DeCodec achieves, for the first time, an codec model that integrates audio reconstruction, SE, background sound extraction, one-shot VC, and can provide noise-robust feature-level controllable support for downstream audio tasks, such as ASR and zero-shot TTS.

## 2 RELATED WORK

**Audio codecs**: Audio codecs were originally used for signal compression (Pan, 1995), and are now widely used as tokenizers for large model approaches due to discretizing and compressing audio features effectively (Défossez et al., 2022; Kumar et al., 2024). In 2021 Zeghidour et al. (2021) proposed SoundStream, which is a novel end-to-end neural audio codec that can efficiently compress speech, music and general audio. It relies on a model architecture composed by a fully convolutional encoder-decoder network and an inserted residual vector quantizer, which forms the basic architecture of the neural codecs. EnCodec (Défossez et al., 2022) introduced a multiscale spectrogram adversary and a loss balancer to mitigate artifacts and enhance sample quality. Concurrently, HiFi-Codec (Yang et al., 2023) employed group-residual vector quantization (GRVQ) to improve reconstruction fidelity while reducing codebook usage. DAC (Kumar et al., 2024) integrates high-fidelity audio generation with image-inspired vector quantization, enabling universal compression of diverse audio domains—such as speech, music, and environmental sounds—within a single model, thus broadening its applicability to generative audio tasks. Different from above, Jiang et al. (2025) proposed UniCodec, a single-codebook model for multi-domain audio (speech, music, sound). It employs a partitioned domain-adaptive codebook and domain-specific Mixture-of-Experts to capture distinct acoustic characteristics. However, its single-codebook design fails to disentangle mixed audio—such as speech with background sound—which is categorized broadly as 'sound', leading to degraded performance in downstream tasks involving real-world recordings.

**Speech Tokenizer**: Currently, research on speech representation is more extensive than in the general audio field. Speech comprises both semantic and paralinguistic information, with different tasks focusing on different aspects (Xu et al., 2023). In response to these diverse requirements, two main categories of speech representation methods have emerged: implicit representations based on self-supervised learning and explicit decoupling methods based on neural codecs. For implicit representation methods in self-supervised learning, models are typically forced to learn high-level representations with linguistic discriminative power through large-scale unsupervised pre-training (Hsu et al., 2021; Baevski et al., 2020; Chen et al., 2022). Although effective for speech recognition, these models entangle semantic and acoustic information in their representations, making them inferior apply to generation tasks (Tsai et al., 2022; Zhang et al., 2023), which require precise control over acoustic properties. For explicit decoupling methods, codec model is typically used as the basic framework, which achieves decoupling of speech components through the design of structured quantizers and supervision strategies (Huang et al., 2023; Du et al., 2024; Liu

et al., 2024). Among them, FACodec (Ju et al., 2024) uses gradient reversal layers to decompose speech into four components: content, pitch, timbre, and acoustic residual. However, there is significant information leakage between the components. In contrast, SpeechTokenizer (Zhang et al., 2023), Mimicodec (Défossez et al., 2024), and DualCodec (Li et al., 2025) adopt a simpler semantic content-acoustic residual decomposition approach. Among these, SpeechTokenizer and Mimicodec utilize HuBERT and WavLM for semantic supervision, respectively, while DualCodec employs a self-supervised strategy. All of them effectively decomposed the speech components. However, these methods are currently only applicable to clean speech input but lack robustness in noise scenarios, limiting their application in actual generation tasks.

# 3 DeCodec

## 3.1 Signal Model and Problem Formulation

Background sound is an additive interference to speech in the time domain, so the mixed signal can be written as

$$\mathbf{y} = \mathbf{s} + \mathbf{n}, \tag{1}$$

where $\mathbf{y}$ denotes the vector forms of mixed signal composed of clean speech $\mathbf{s}$ and background sound $\mathbf{n}$. Based on the differences in the physical generation mechanisms, it can be assumed that speech signal and background sound signal are mutually independent (Zheng et al., 2023). Given this, an audio feature extractor can theoretically transform the mixed signal into an embedded space and provide decoupled representations of speech and background sound.

## 3.2 System Overview

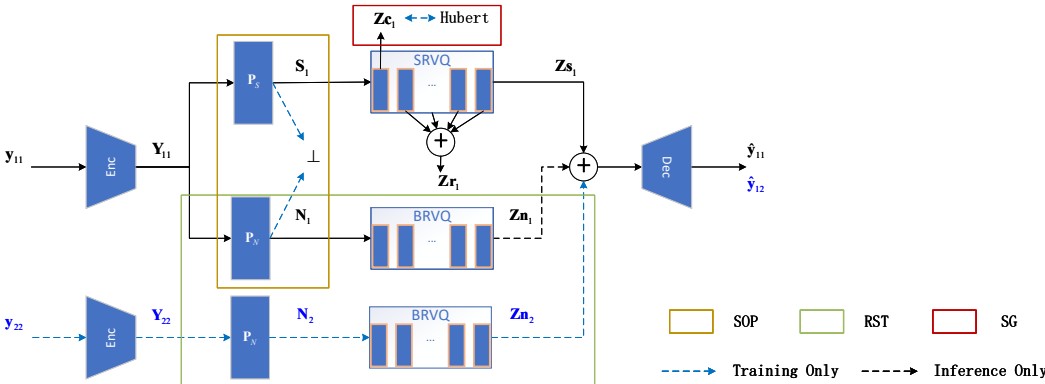

Figure 2: The overview of the proposed DeCodec.

The proposed DeCodec consists of four main components: an encoder, a SOP module, a parallel residual vector quantizers (RVQs) with SG, and a decoder, as shown in Figure 2. The encoder is used to preliminarily convert the time-domain signal $\mathbf{y}$ into an embedded representation $\mathbf{Y}$ based on the physical properties of the audio. The SOP module simulates the left and right hemispheres of the A2 in the brain, using subspace orthogonal projection to enable each hemisphere to process distinct information, *i.e.* $\mathbf{S}$ and $\mathbf{N}$, within the audio. The parallel RVQs simulates the specific processing in A2, where SRVQ simulates the processing of speech information by the left hemisphere, yielding the speech quantization vector $\mathbf{Zs}$; while NRVQ simulates the processing of background sound by the right hemisphere, yielding the background sound quantization vector $\mathbf{Zn}$. Furthermore, for speech information, the SG in the SRVQ can decompose it into a semantic quantization vector $\mathbf{Zc}$ and a residual paralinguistic quantization vector $\mathbf{Zr}$ which contain paralinguistic information. The decoder corresponds to the motor cortex, decoding the above quantization vectors back into time-domain audio $\hat{\mathbf{y}}$.

### 3.3 ENCODER AND DECODER

The convolutional-based encoder-decoder network from DAC (Kumar et al., 2024) is adopted in the proposed DeCodec. In particular, the encoder consists of a 1D convolution with channel of $C$ and kernel size of $K$ and $M$ encoder blocks adopted in DAC, which performs temporal downscaling with a chosen striding factor. Notably, we provide both causal/non-causal versions of the encoder. For the causal version, two-layer LSTM is employed after the encoder blocks for time sequence modeling, while for the non-causal version, two-layer BiLSTM is adopted instead. Finally, there is a final 1D convolution layer with a kernel size of $K$ and an output channel count of $D$.

The decoder mirrors the encoder, using transposed convolutions instead of strided convolutions, and with the strides in reverse order as in the encoder, outputting the final reconstructed audio.

### 3.4 SOP MODULE

Inspired by the cerebral cortical processing of speech and background sound in separate regions, we propose that the primary representation $\mathbf{Y}$ of the input mixture $\mathbf{y}$ can also be further decomposed into speech representation $\mathbf{S}$ and background sound representation $\mathbf{N}$. Formally, for every $\mathbf{Y} \in \mathcal{V}_{\mathcal{Y}}$, there exits $\mathbf{S} \in \mathcal{V}_{\mathcal{S}}$ and $\mathbf{N} \in \mathcal{V}_{\mathcal{N}}$, such that

$$\mathbf{Y} = \mathbf{S} + \mathbf{N}, \tag{2}$$

where $\mathcal{V}_{\mathcal{Y}}$ is the space for mixture embeddings, and $\mathcal{V}_{\mathcal{S}}$ and $\mathcal{V}_{\mathcal{N}}$ are subspaces representing speech and background sound, respectively. This decomposition satisfies:

$$\mathcal{V}_{\mathcal{Y}} = \mathrm{span}(\mathcal{V}_S \cup \mathcal{V}_N), \quad \mathcal{V}_S \cap \mathcal{V}_N = \mathbf{0}, \tag{3}$$

equivalently, $\mathcal{V}_Y = \mathcal{V}_S \oplus \mathcal{V}_N$. Let $\mathbf{P}_S : \mathcal{V}_Y \to \mathcal{V}_S$ and $\mathbf{P}_N : \mathcal{V}_Y \to \mathcal{V}_N$ be orthogonal projection operators. The embedding $\mathbf{Y}$ admits the orthogonal decomposition:

$$\mathbf{Y} = \mathbf{S} + \mathbf{N} = \mathbf{P}_S \mathbf{Y} + \mathbf{P}_N \mathbf{Y}, \tag{4}$$

where $\mathbf{S} \perp \mathbf{N}$ (i.e., $\langle \mathbf{S}, \mathbf{N} \rangle = 0$), and the projectors satisfy $\mathbf{P}_S + \mathbf{P}_N = \mathbf{I}$ with $\mathbf{P}_S \mathbf{P}_N^T = \mathbf{0}$.

Therefore, we introduce two trainable linear projection layers followed by the encoder to model $\mathbf{P}_S$ and $\mathbf{P}_N$, respectively, as shown in the yellow box of Figure 2. To enforce orthogonality between $\mathbf{P}_S$ and $\mathbf{P}_N$, we impose the following constraint on their outputs $\mathbf{S}$ and $\mathbf{N}$:

$$\mathcal{L}_\perp = \|\langle \mathbf{S}, \mathbf{N} \rangle - \mathbf{0}\|_2, \tag{5}$$

where $\|\cdot\|_2$ is the $L^2$ norm, and $\mathcal{L}_\perp$ is minimized to ensure that $\mathbf{S}$ and $\mathbf{N}$ are mutually orthogonal. From this orthogonality constraint $\mathcal{L}_\perp$ and Equation (4), the following relationship can be derived:

$$\mathbf{S}\mathbf{N}^T = (\mathbf{P}_S \mathbf{Y})(\mathbf{P}_N \mathbf{Y})^T = \mathbf{P}_S \mathbf{Y} \mathbf{Y}^T \mathbf{P}_N^T. \tag{6}$$

When the covariance matrix $\mathbf{Y}\mathbf{Y}^T$ satisfies the angular matrix, indicating that the encoder extracts sufficiently diverse embeddings with different feature channels being mutually independent, we can obtain $\mathbf{P}_S \mathbf{P}_N^T = \mathbf{0}$, which demonstrates that $\mathbf{P}_S$ and $\mathbf{P}_N$ are indeed orthogonal projection matrices. And thus our SOP methods ensures the subspaces spanned by $\mathbf{P}_S$ and $\mathbf{P}_N$ to be disentangled, thereby promoting a complete decoupling between speech and background sound representations.

### 3.5 PARALLEL RVQs WITH SG

After passing through the SOP module, the mixed embeddings $\mathbf{Y}$ is decoupled into two continuous representations $\mathbf{S}$ and $\mathbf{N}$. To discrete the above representations into corresponding quantized vector, we involved parallel RVQs for DeCodec, where SRVQ and NRVQ are used to quantize $\mathbf{S}$ andn$\mathbf{N}$, respectively. Specifically, both SRVQ and NRVQ use residual vector quantization (Vasuki & Vanathi, 2006). The discrete speech representation can be obtained by $\mathbf{Zs} = \sum_{k=1}^{K_s} \mathbf{Qs}^k$ where $K_s$ donates number of vector quantizers, $\mathbf{Qs}^k$ is the quantized speech vector obtained by the $k$-th quantizer. Likewise, the discrete background sound representation can be expressed as $\mathbf{Zn} = \sum_{k=1}^{K_n} \mathbf{Qn}^k$.

Moreover, for SRVQ, as semantic guidance has been proven effective, we have also applied base 960h Hubert-L9 [1] as semantic guidance in the first layer of the SRVQ quantizer, as shown in the red box of Figure 2. The SG is expected to further help SRVQ decompose speech into semantic representation $\mathbf{Zc} = \mathbf{Qs}^1$ and residual paralinguistic representation $\mathbf{Zr} = \sum_{k=2}^{K_s}$. And the SG constraint (Zhang et al., 2023) can be expressed as:

$$\mathcal{L}_{\text{SG}} = \|log\sigma\left(\cos(\mathbf{WZc}, \mathcal{H})\right)\|_1, \tag{7}$$

where $\mathbf{Zc}$ and $\mathcal{H}$ denote the quantized output of the first layer of SRVQ and Hubert-L9 representation of the corresponding clean speech $\mathbf{s}$, respectively. $\mathbf{W}$ denotes the projection matrix that projects $\mathbf{Zc}$ onto the same feature dimension as $\mathcal{H}$. $\cos(\cdot)$ represents cosine similarity along the frame dimension and $\sigma(\cdot)$ denotes sigmoid activation. $\|\cdot\|_1$ denotes the $L^1$ norm of the feature dimension.

### 3.6 RST procedure

Inspired by neural developmental feedback mechanisms, we involved a novel representation swap training procedure that enforces disentanglement between speech and background sound and guides $\mathbf{S}$ and $\mathbf{N}$ to be speech representations and background sound representations, respectively. The procedure is shown in Figure 2. Given two uncorrelated mixed speech samples $\mathbf{y}_{11} = \mathbf{s}_1 + \mathbf{n}_1$ and $\mathbf{y}_{22} = \mathbf{s}_2 + \mathbf{n}_2$, the RST proceeds as follows:

First, the encoder processes both samples to extract corresponding continuous embeddings $\mathbf{Y}_{11}, \mathbf{Y}_{22}$, respectively. These embeddings are then decoupled into orthogonal subspaces by the SOP module and we only take the speech representation $\mathbf{S}_1$ of $\mathbf{Y}_{11}$ and the background sound representation $\mathbf{N}_2$ of $\mathbf{Y}_{22}$. Then the quantinized vector of $\mathbf{S}_1$ and $\mathbf{N}_2$ are obtained by SRVQ and NRVQ of the parallel RVQ module, respectively. Finally, the reconstructed hybrid signal $\hat{\mathbf{y}}_{12}$ is obtained by the decoder. The entire process can be summarized as follows:

$$\mathbf{Y}_{11} = \text{Enc}(\mathbf{y}_{11}), \mathbf{Y}_{22} = \text{Enc}(\mathbf{y}_{22}). \tag{8}$$
$$\mathbf{S}_1 = \mathbf{P}_S\mathbf{Y}_{11}, \mathbf{N}_2 = \mathbf{P}_N\mathbf{Y}_{22}. \tag{9}$$
$$\mathbf{Zs}_1 = \text{SRVQ}(\mathbf{S}_1), \mathbf{Zn}_2 = \text{NRVQ}(\mathbf{N}_2). \tag{10}$$
$$\hat{\mathbf{y}}_{12} = \text{Dec}(\mathbf{Zs}_1 + \mathbf{Zn}_2) \approx \mathbf{s}_1 + \mathbf{n}_2. \tag{11}$$

Correspondingly, we designed the RST loss to guide the proposed system to achieve decoupling of speech and background sound representation. The RST loss is given as:

$$\mathcal{L}_{\text{RST}} = \|\text{Dec}(\mathbf{Zs}_1 + \mathbf{Zn}_2) - (\mathbf{s}_1 + \mathbf{n}_2)\|_1. \tag{12}$$

According to Section 3.4, the orthogonal constraint $\mathcal{L}_\perp$ ensures that the subspaces remain statistically independent throughout the training process. Here, we theoretically prove that the proposed $\mathcal{L}_{\text{RST}}$ can further force $\mathbf{Zs} \in \mathcal{V}_S$ to be speech representations only, while $\mathbf{Zn} \in \mathcal{V}_N$ to be background sound representations only.

Given $\mathbf{Zs}_1 + \mathbf{Zn}_1$ and $\mathbf{Zs}_1 + \mathbf{Zn}_2$, SOP Loss minimization requires:

$$\text{Dec}(\mathbf{Zs}_1 + \mathbf{Zn}_1) = \hat{\mathbf{s}}_1 + \hat{\mathbf{n}}_1 \approx \mathbf{s}_1 + \mathbf{n}_1, \tag{13}$$
$$\text{Dec}(\mathbf{Zs}_1 + \mathbf{Zn}_2) = \hat{\mathbf{s}}_1 + \hat{\mathbf{n}}_2 \approx \mathbf{s}_1 + \mathbf{n}_2. \tag{14}$$

Subtracting Equation (13) from Equation (14) can obtain:

$$\text{Dec}(\mathbf{Zs}_1 + \mathbf{Zn}_2) - \text{Dec}(\mathbf{Zs}_1 + \mathbf{Zn}_1) \approx \mathbf{n}_2 - \mathbf{n}_1. \tag{15}$$

By the mean value theorem for vector functions (Russell, 2020), there exists $\xi$ between $\mathbf{Zn}_1$ and $\mathbf{Zn}_2$ such that:

$$\frac{\partial \text{Dec}}{\partial \mathbf{Zn}}\Big|_\xi (\mathbf{Zn}_2 - \mathbf{Zn}_1) \approx \mathbf{n}_2 - \mathbf{n}_1. \tag{16}$$

The left side depends on $\mathbf{Zs}_1$ through $\xi$, while the right side is independent of $\mathbf{Zs}_1$. Therefore, for consistency $\forall \mathbf{n}_1, \mathbf{n}_2$, $\mathbf{Zs}_1$ must be independent of $\mathbf{n}_1$, which means that the quantinized speech vector $\mathbf{Zs}$ does not contain background sound information. The same can be proven for the quantinized background sound vector $\mathbf{Zn}$, which does not contain speech information.

---

[1]https://huggingface.co/collections/facebook/hubert-651fca95d57549832161e6b6

## 4 EXPERIMENTS

### 4.1 EXPERIMENTAL SETUP

**Datasets:** For training, multilingual datasets were used at 16kHz sample rate, including Aishell3 (Shi et al., 2020), train-clean-100&360 of LibriTTS (Zen et al., 2019), VCTK (Veaux et al., 2013), WSJ1 and WSJ0 (Corpus, 1992). Aound 700h of speech data were randomly selected and mixed with randomly selected background sound from ESC-50 (Piczak, 2015) and DNS-Noise (Reddy et al., 2020) at random signal-to-noise ratio (SNR) ranging from -5 to 40 dB to form the training set. For evaluation, the LibriSpeech Test-clean set served as the clean speech test set. A noisy speech test set was created by mixing Test-clean with DNS-Noise at SNRs from -5 to 20 dB. 300 clips were randomly selected from each set. Besides, to compare with the SOTA SE methods, the public 2nd DNS Challenge (Reddy et al., 2020) blind test sets is used for speech enhancement evaluation.

**Evaluation:** Signal-to-distortion rate (SDR) (Vincent et al., 2007), Mel distance and word error rate (WER) based on Whisper ASR model (Radford et al., 2023) are employed for signal reconstruction evaluation. Non-intrusive p.835 DNSMOS scores (Reddy et al., 2021) are used to evaluate the SE performance. WER and speaker similarity (SIM) calculated by WavLM-TDNN (Chen et al., 2022) are adopted for evaluating one-shot VC.

**Baseline models:** For reconstruction, Four strong codec baselines are adopted, namely EnCodec (Défossez et al., 2022)[2], HiFi-Codec (Yang et al., 2023)[3], DAC (Kumar et al., 2024)[4], and SpeechTokenizer (Zhang et al., 2023)[5]. All these models are infered from offical checkpoints. For SE, Three types of advancing SE baselines are adopted namely Inter-SubNet (Chen et al., 2023), StoRM (Lemercier et al., 2023)[6] and SELM (Wang et al., 2024). The results of these models on the DNS Challenge testset are taken from the paper (Wang et al., 2024). For one-shot VC, the advanced SpeechTokenizer is used as baseline. Further, to compare the cascated pipline with the proposed DeCodec, the advanced StoRM is selected as SpeechTokenizer's denoising front-end, denoted as StoRM-SpeechTokenizer.

### 4.2 EXPERIMENTAL RESULTS

We conduct two sets of experiments to thoroughly evaluate the performance of the proposed DeCodec. The performance of audio processing tasks, which DeCodec can achieve on its own, are shown in this section. Further, the performance of DeCodec as a feature extractor for downstream ASR and TTS models are shown in Appendix F and G, respectively.

#### 4.2.1 CODEC

Table 1: Reconstruction performance of codec models.

| Codec Models | kpbs | Causal | Clean | | | Noisy | |
|---|---|---|---|---|---|---|---|
| | | | SDR↑ | Mel Distance↓ | WER↓ | SDR↑ | Mel Distance↓ |
| EnCodec | 6.0 | ✓ | 6.86 | 1.03 | 2.28 | 4.88 | 0.84 |
| HiFi-Codec | 2.0 | ✓ | 4.85 | 0.75 | 2.61 | -0.66 | 0.90 |
| DAC | 4.5 | ✓ | 0.60 | **0.65** | 2.21 | -1.62 | **0.69** |
| SpeechTokenizer | 4.0 | ✗ | 3.41 | 0.76 | **1.82** | -0.50 | 0.90 |
| DeCodec-c | 4.0+4.0 | ✓ | 6.79 | 0.88 | 1.98 | 4.62 | 0.82 |
| DeCodec | 4.0+4.0 | ✗ | **7.61** | 0.89 | 1.92 | **5.21** | 0.81 |

Table 1 demonstrates the audio reconstruction performance of the DeCodec and baseline codec models. As shown in the Table 1, regardless of whether it is clean or noisy speech, the proposed DeCodec achieves the highest SDR for speech reconstruction, indicating the lowest temporal distortion. Additionally, the causal DeCodec achieves an SDR comparable to that of the EnCodec. These results demonstrate that the proposed system can ensure the

---

[2]https://github.com/facebookresearch/encodec

[3]https://github.com/yangdongchao/AcademiCodec

[4]https://github.com/descriptinc/descript-audio-codec

[5]https://github.com/ZhangXInFD/SpeechTokenizer

[6]https://github.com/sp-uhh/storm

performance of complete signal reconstruction while decoupling representations. From the perspective of semantic preservation, the WER of the reconstructed speech by the proposed DeCodec is only slightly worse than that of SpeechTokenizer but significantly better than that of the baseline codes trained without semantic guidance, further confirming the role of semantic guidance in reconstructing semantic information. Additionally, in terms of mel distance, while the proposed algorithm does not show a significant advantage in clean speech, it outperforms SpeechTokenizer and other baselines in noisy scenarios, with performance second only to DAC, demonstrating the advantage of the DeCodec over other codecs in reconstructing background sound. Overall, the proposed DeCodec performs comparably to existing codec models in reconstruction while possessing its own distinctive advantages.

### 4.2.2 SE

Speech enhancement for noisy speech can be easily achieved by decoding the recombined reprezentaions of the DeCodec, which replace the background sound reprezentaions of the input noisy speech with the background sound reprezentaions of a blank audio with the same length. Table 2 presents the speech enhancement performance of the proposed DeCodec compared with different tpyes of SE baselines. As shown in the Table 2, the proposed DeCodec achieved the highest DNSMOS scores in both simulation and real recording test sets, demonstrating that it is effective in decoupling speech from background sound and providing a new approach for speech enhancement. Specifically, as the BAK scores show, the proposed DeCodec outperforms various existing SE models in background sound suppression, indicating that decoupling in the representation domain can sufficiently control the retention or removal of background sound. The speech signal distortion of the proposed DeCodec, as shown by the SIG score, is slightly inferior to SELM in real recordings, possibly because the proposed method, compared to a single SE model, includes discretization quantizers, resulting in slightly inferior speech signal reconstruction compared to SE models that contain complete continuous speech information. Additionally, the proposed causal DeCodec model achieves performance comparable to the non-causal SELM model and significantly outperforms the causal Inter-SubNet model. This suggests that the proposed speech-background sound representation decoupling method holds significant implications for the design of future causal speech enhancement models.

Table 2: The DNSMOS scores on different SE models on the DNS Challenge test set.

| SE Models | Model Type | Causal | Without Reverb | | | Real Recordings | | |
|---|---|---|---|---|---|---|---|---|
| | | | OVL↑ | SIG↑ | BAK↑ | OVL↑ | SIG↑ | BAK↑ |
| Noisy | - | - | 2.48 | 3.39 | 2.62 | 2.26 | 3.05 | 2.51 |
| Inter-SubNet | Discriminative | ✓ | 3.10 | 3.46 | 3.82 | 2.81 | 3.26 | 3.57 |
| StoRM | Diffusion | × | 3.21 | 3.51 | 3.94 | 2.94 | 3.41 | 3.38 |
| SELM | Transformer | × | 3.26 | 3.51 | 4.10 | 3.12 | **3.59** | 3.44 |
| DeCodec-c | Codec | ✓ | 3.31 | 3.58 | 4.09 | 2.99 | 3.31 | 3.94 |
| DeCodec | Codec | × | **3.39** | **3.64** | **4.13** | **3.13** | 3.45 | **3.99** |

### 4.2.3 ONE-SHOT VC

To demonstrate that semantic and paralinguistic information can be hierarchically represented during the quantization process by the proposed DeCodec while decoupling speech and background sound, we conduct one-shot VC experiment on noisy speech test set by decoding the recombined representaions of the DeCodec, which replacing the SRVQ-2:8 of the input noisy speech with the SRVQ-2:8 of the reference noisy speech. The reference speech is processed to match the input speech length.

Results of one-shot VC on different codec models on the noisy speech test set is shown in Table 3. Note that in order to evaluate the semantic retention of the speech after one-shot VC, DeCodec removed background sound by the SE method as in Section 4.2.2 when performing voice conversion, *i.e.*, one-shot VC+SE. Correspondingly, since SpeechTokenizer is not robust to noise, for fairness, we also cascaded the optimal StoRM denoising model in Section 4.2.2 as its front end. Table 3 shows that existing speech entangled codec models such as SpeechTokenizer is not robust to noisy speech, as the WER of the converted speech reaches 74.18, indicating the speech is unintelligible. However, the proposed DeCodec and

Table 3: Results of one-shot VC on different codec models on the noisy speech test set.

| Codec Models | Source Speech | Reference Speech | Output BGS | SIM↑ | WER↓ |
|---|---|---|---|---|---|
| Reference | - | - | - | 0.69 | - |
| SpeechTokenizer | RVQ-1 | RVQ-2:8 | - | 0.80 | 74.18 |
| StoRM-SpeechTokenizer | RVQ-1 | RVQ-2:8 | - | **0.83** | 52.73 |
| DeCodec | SRVQ-1 | SRVQ-2:8 | × BRVQ-1:8 | **0.83** | **50.46** |

SpeechTokenizer after front-end StoRM denoising can effectively reduce the WER of the converted speech, and the speaker similarity is also improved. This confirms that the proposed algorithm can decompose semantic and paralinguistic information while being robust to noise. As for the relatively high WER, it may be due to the different speech segment voicing times. When the input speech is voiced but the reference speech is not, the voice tone cannot be effectively converted by only switching representations, resulting in high distortion of the converted speech and a noticeable increase in WER. Besides, the proposed DeCodec achieves a lower WER compared to the StoRM-SpeechTokenizer method, indicating that the proposed approach of decoupling speech and background sound in the representation domain introduces less error than the front-end time-domain separation method, and thus has a competitive advantage in information control for audio tasks.

### 4.2.4  ABLATION STUDY

Table 4: Results of ablation studies on Decodec on the noisy speech test set.

| | Module | | | Causal | Overall | Decoupling | | |
|---|---|---|---|---|---|---|---|---|
| | SOP | RST | SG | | SDR-O↑ | SDR-B↑ | SDR-S↑ | WER*↓ |
| Ablation-1 | ✓ | - | - | ✓ | **8.93** | -13.15 | -1.91 | - |
| Ablation-2 | - | ✓ | - | ✓ | 6.70 | -10.67 | 3.03 | - |
| Ablation-3 | ✓ | ✓ | - | ✓ | 6.68 | **0.49** | **7.90** | 41.9 |
| DeCodec-c | ✓ | ✓ | ✓ | ✓ | 4.62 | -1.11 | 5.70 | 25.8 |
| DeCodec | ✓ | ✓ | ✓ | × | 5.21 | -0.36 | 6.73 | **23.6** |

To validate the effectiveness of the proposed SOP vlock, RST procedure, and SG method, we conducted ablation studies, with the results shown in Table 4. SDR-O, SDR-B, and SDR-S represent the signal distortion ratios for reconstructing the original audio, decoupled background sound, and decoupled speech, respectively. WER* is the WER of the downstream ASR model, indicating the ability of the codec used in training the downstream ASR model to represent semantic information. The SDR-B values for Ablation-1 and Ablation-2 are both below -10 dB, and the SDR-S values are also low, indicating that relying solely on a single SOP module or RST procedure is insufficient to achieve effective decoupling of speech and background sound. However, the results of Ablation-3 show significant improvements in both SDR-B and SDR-S, indicating that the joint use of the proposed SOP module and RST procedure can effectively decouple speech and background sound in the representation domain. On the basis of Ablation-3, DeCodec further introduces SG, resulting in a slight decrease in SDR but a significant reduction in WER*. This indicates that with the proposed SOP+RST method for achieving speech-background sound decoupling, the introduction of a speech decomposition mechanism can further enable hierarchical representation of semantic and paralinguistic information, thereby expanding the scope of audio processing applications for the proposed DeCodec. Additionally, compared to the causal version, the non-causal version of DeCodec demonstrates improved performance, allowing users to select the appropriate version based on specific audio processing requirements.

## 5  CONCLUSION

This work presents DeCodec, reframing audio codecs as an universal disentangled representation learner to achieve hierarchical disentanglement for representing speech-background sound and semantic-paralinguistic. The experimental results confirm that the representations are sufficiently disentangled, enabling controllable feature selection tailored to diverse downstream tasks. Limitations and future directions are discussed in Appendix H.

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

## A    Total Training Loss

**Loss functions:** DeCodec is trained based on GAN framework. In addition to $\mathcal{L}_\perp$, $\mathcal{L}_{SG}$ and $\mathcal{L}_{RST}$ mentioned in Section 3, multi-scale mel-spectral loss (Kumar et al., 2024) to $\hat{\mathbf{y}}_{12}$ is used as ***reconstruction loss***. ***Discriminative loss*** (Yang et al., 2023) comprises multi-period discriminator loss and multi-band multi-scale STFT discriminator loss. Original codebook and commitment losses (Van Den Oord et al., 2017) to both SRVQ and NRVQ are used as ***RVQ loss***.

**Loss weights:** The weights of each loss are as follows: 500.0 for $\mathcal{L}_{RST}$, 150.0 for $\mathcal{L}_{SG}$, 10.0 for $\mathcal{L}_{SG}$, 10.0 for ***reconstruction loss***, 1.0 for ***discriminative loss***, and 1.0, 10.0 for codebook and commitment losses in ***RVQ loss*** for the total parallel RVQs, respectively.

## B    Model Configuration

The Encoder of the DeCodec cascades of 4 encoder blocks used in DAC, downsampling the input audio waveform at rates [2, 4, 5, 8]. The $C$, $D$, $k$ of the corresponding 1D convolution are 32, 1024, 7, respectively. The Decoder has 4 corresponding decoder blocks, upsampling at rates [8, 5, 4, 2]. The decoder dimension is 1536. $\mathbf{P}_S$ and $\mathbf{P}_N$ are both set to project onto a 1024-dimensional subspace. The number of quantizers for SRVQ and NRVQ are both 8, and the codebook size are both 1024.

The DeCodec is trained with a batch size of 12 for 1000k iterations. The AdamW optimizer is used with a learning rate of $10^{-4}$, which decays at every step with $\gamma = 0.999996$.

## C    Guidlines for DeCodec

Table 5: Guidelines for using DeCodec to perform speech reconstruction, SE, background sound extraction, one-shot VC, and one-shot VC+SE functions.

| Task | Input Audio | | | Reference Audio | | | Blank Audio | | |
|---|---|---|---|---|---|---|---|---|---|
| | SRVQ-1/Zc | SRVQ-2:8/Zr | BRVQ-1:8/Zb | SRVQ-1/Zc | SRVQ-2:8/Zr | BRVQ-1:8/Zb | SRVQ-1/Zc | SRVQ-2:8/Zr | BRVQ-1:8/Zb |
| Reconstruction | ✓ | ✓ | ✓ | - | - | - | - | - | - |
| SE | ✓ | ✓ | - | - | - | - | - | - | ✓ |
| BGS Extraction | - | - | ✓ | - | - | - | ✓ | ✓ | - |
| one-shot VC | ✓ | - | ✓ | - | ✓ | - | - | - | - |
| one-shot VC+SE | ✓ | - | - | - | ✓ | - | - | - | ✓ |

Since DeCodec is based on the Codec framework and aims to achieve decoupling of speech and background sound representations, as well as further decomposition of speech into semantic and residual sublinguistic representations, it can theoretically achieve audio processing tasks such as audio coding reconstruction, speech enhancement (denoising), background sound extraction, and one-shot voice conversion with controllable background sound through explicit combination of representations. The guidelines for different tasks and their corresponding representation combinations are shown in Table 5.

## D    Component Decoupling Analysis

We plot Figure 3 to provide a more intuitive analysis of the subspace projection weights of the proposed SOP module. Figure 3 (a) plots the distribution of cosine similarity between $\mathbf{P}_S$ and $\mathbf{P}_N$, and Figure 3 (b) plots the singular value comparision between $\mathbf{P}_S$ and $\mathbf{P}_N$. In Figure 3 (a), the cosine similarity distribution between $\mathbf{P}_S$ and $\mathbf{P}_N$ approaches 0 and follows a normal distribution with a mean of 0. In Figure 3 (b), the singular values of $\mathbf{P}_S$ and $\mathbf{P}_N$ are also distinct, confirming that the proposed SOP module indeed achieves a subspace orthogonal decomposition.

As for the validation of the SG module, we plotted the semantic representation $\mathbf{Zc}$ and residual paralinguistic representation $\mathbf{Zr}$ of SRVQ, with Figure 6(a) corresponding to $\mathbf{Zc}$ and Figure 6(b) corresponding to $\mathbf{Zr}$. Specifically, We randomly select five speakers from the noisy test set and pick 5 random speech samples per speaker. By performing mean pooling to $\mathbf{Zc}$ and $\mathbf{Zr}$ along the temporal dimension, each representation is converted into

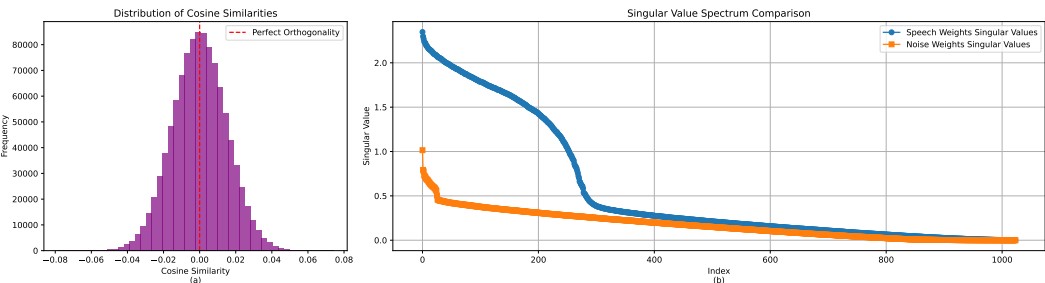

Figure 3: Analysis of the proposed SOP module: (a) distribution of cosine similarity between $\mathbf{P}_S$ and $\mathbf{P}_N$, and (b) singular value comparision between $\mathbf{P}_S$ and $\mathbf{P}_N$.

a single vector. These vectors are then visualized in a 2D space using t-SNE, with speech samples from the same speaker represented in the same color. From Figure 4 (a), it can be observed that $\mathbf{Zc}$ for different speakers are scattered randomly without discernible pattern. In contrast, the $\mathbf{Zr}$ for the same speaker tend to cluster together in Figure 4 (b), while being distinct from those of other speakers. This indicates that all speaker-related paralinguistic information is decomposed into $\mathbf{Zr}$, while $\mathbf{Zc}$ is unrelated to the speaker's information.

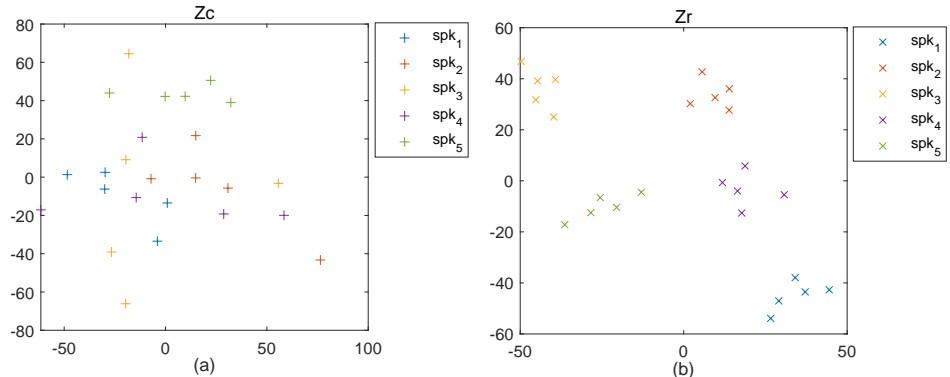

Figure 4: Visualization of $\mathbf{Zc}$ and $\mathbf{Zr}$ of Decodec: (a) $\mathbf{Zc}$, and (b) $\mathbf{Zr}$.

## E  SPECTRAL ANALYSIS

Figure 2 provide a more intuitive illustration of two demos processed by DeCodec on various audio processing tasks.

Figure 2 (b-1) and (b-2) provide show the reconstruction performance of DeCodec. Compared with the original inputs shown in Figure 2 (a-1) and (a-2), it is evident the proposed DeCodec performs well in terms of both formant reconstruction and background sound detail preserved.

Figure 2 (c-1) and (c-2) demonstrate the validity of the proposed DeCodec for speech enhancement. Compared with the original inputs shown in Figure 2 (a-1) and (a-2), the background sound of the enhanced speech is significantly suppressed without noticeable residual, while the remained speech retains clear resonance peaks without significant magnitude distortion. In addition, the opposite operation of speech enhancement can be used to extract background sound. As can be seen in Figure 2 (d-1) and (d-2), the extracted background sounds do not contain residual speech as well. These confirm that the proposed DeCodec does indeed effectively decouple speech and background sound in the representation domain.

Figures 2 (f-1) and (f-2) show the results of swapping the voice tones of Figures 2 (a-1) and (a-2) with background sound removed. As can be clearly seen that after the voice tone is

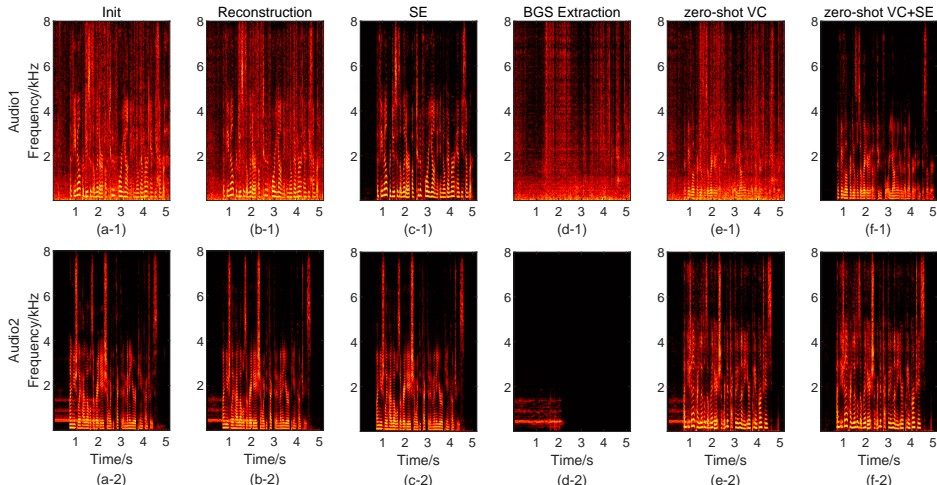

Figure 5: Demos of audio tasks processed by DeCodec, with the first row for audio demo 1 and the second row for audio demo 2.

swapped, the fundamental frequency (F0) of Figures 2 (f-1) decreases, resembling that of Figures 2 (a-2); while the F0 of Figures 2 (f-2) increases, resembling that of Figures 2 (a-1), indicating that the voice has been effectively converted. As for the content, the spectral distribution of Figures 2 (f-1) remains similar to that of Figures 2 (a-1), and so does Figures 2 (f-2) to that of Figures 2 (a-2), indicating that the content remains unchanged. The above results confirm that the proposed DeCodec can achieve decompositional representation of semantic-paralinguistic information.

Additionally, as shown in Figures 2 (e-1) and (e-2), the proposed DeCodec can also retain the background sound of the input speech, ensuring that the converted speech remains in the same acoustic environment, thereby enhancing the realism.

## F  Performance on Downstream ASR

We adopt a Decoder-only Transformer as the downstream ASR model and use only Librispeech for training to verify the performance of the proposed DeCodec as a feature extractor. Other SpeechTokenizer-based feature extractors are also evaluated to compared the performance. The WER* results of ASR models trained with different feature extractors on clean speech test sets and noisy speech test sets are shown in Table 6. As can be seen, compared to DAC which has no semantic guidance, the WER* of SpeechTokenizer and the proposed DeCodec are significantly reduced, confirming the necessity of SG for downstream ASR tasks. However, for noisy speech, the WER* of SpeechTokenizer significantly increases, primarily due to its lack of robustness to noisy speech. To employ SpeechTokenizer in noisy environments, the conventional approach involves cascading a speech separation model as its front-end. To address this, we utilize StoRM from Section 4.2.2 as the front-end for speech enhancement. Experimental results show that the StoRM-SpeechTokenizer approach effectively reduces WER* in noisy speech compared to using SpeechTokenizer alone, but it is still significantly higher than the proposed DeCodec.This may stem from error accumulation from the StoRM into the ASR model, coupled with the downstream ASR model not being fine-tuned on StoRM-enhanced speech. In contrast, the proposed DeCodec avoids these drawbacks by decoupling speech and background sound in the representation domain, thereby enhancing the robustness of the ASR model against noise.

Furthermore, compared with only using semantic representations (SRVQ-1), the WER decreases by approximately 3 compared to using full speech representations (SRVQ-1:8). This confirms that the vast majority of semantic information is concentrated in the semantic representations, and thus the proposed DeCodec achieves the decomposition of semantic and paralinguistic information.

Table 6: The WER* results of ASR based on different codec models.

| Codec Models | ASR Model | Clean | | Noisy | |
|---|---|---|---|---|---|
| | | (S)RVQ-1 | (S)RVQ-1:8 | (S)RVQ-1 | (S)RVQ-1:8 |
| Hubert-L9 | | *5.8* | | *12.4* | |
| DAC | Decoder-only Transformer | 93.8 | 35.1 | 78.0 | 82.5 |
| SpeechTokenizer | | 15.5 | 13.5 | 59.2 | 55.1 |
| StoRM-SpeechTokenizer | | 15.5 | 13.7 | 34.5 | 32.1 |
| DeCodec | | **14.7** | **12.5** | **26.7** | **23.6** |

# G  PERFORMANCE ON DOWNSTREAM ZERO-SHOT TTS

We adopt VALL-E (Wang et al., 2023a)[7] as the downstream TTS model and use only Librispeech for training to verify the performance of the proposed DeCodec as a feature extractor. Other SpeechTokenizer-based feature extractors are also evaluated to compared the performance.

Since our testing primarily focuses on real-world input speech which contains background sound, objective metrics like WER may be impacted by it. Therefore, we directly invited 10 professional volunteers to conduct subjective evaluations. We determine the Mean Opinion Score (MOS), Similarity Mean Opinion Score (SMOS), Background sound Removal Mean Opinion Score (BRMOS), and Background sound Preservation Mean Opinion Score (BP-MOS) through subjective evaluations, each of which span from 1 to 5. MOS reflects the naturalness of overall generated speech. For generated speech without background sound, SMOS assesses the degree of similarity to the original speaker's voice while BRMOS reflects the removal level of background sound in the generated speech. For generated speech with background sound preserved, BPMOS indicates the quality of the retained background sound. The experimental results are shown in Table 7

Table 7: The subjective results of zero-shot TTS based on different codecs on the noisy speech test set.

| Codec Models | TTS Model | Output BGS | MOS↑ | SMOS↑ | BRMOS↑ | BPMOS↑ |
|---|---|---|---|---|---|---|
| SpeechTokenizer | | - | 1.48 | 1.82 | 1.97 | - |
| StoRM-SpeechTokenizer | VALL-E | - | **4.05** | 3.68 | 4.68 | - |
| DeCodec | | ✓ BRVQ-1:8 | 3.96 | **3.69** | **4.74** | - |
| DeCodec | | ✗ BRVQ-1:8 | 4.09 | - | - | 4.19 |

As shown in 7, the SpeechTokenizer, which trained solely for speech coding, suffers a significant performance loss when processing noisy speech, generating poor-quality speech with significant residual background sound. By incorporating a SE front-end into the SpeechTokenizer, its performance on noisy speech inputs is significantly improved. In contrast, the proposed DeCodec achieves comparable overall performance to StoRM-SpeechTokenizer without requiring a SE front-end, only leveraging decoupled representations. Notably, the VALLE model was trained exclusively on clean speech during training. And as shown in Table 7, the proposed DeCodec outperforms traditional cascaded approaches in background sound removal, achieving a BRMOS score of 4.74. This confirms that the proposed speech-background sound representation decoupling can significantly enhance the robustness of downstream model without additional fine-tuning. The SMOS in Table 7 also demonstrates that the proposed DeCodec maintains the ability of the hierarchical speech codecs to decompose semantic and paralinguistic information while decoupling speech and background sound representations.

Furthermore, the proposed DeCodec can directly control whether generated speech retains the background sound of the input speech by preserving or discarding the background sound representation. As shown in 7, the synthesized speech by DeCodec with retained background sound achieves a MOS score of 4.09, demonstrating higher realism.

---

[7]https://github.com/lifeiteng/vall-e

## H    Limitation and Future work

The proposed DeCodec currently aims to decoupling speech and background sound represen­tations of input audio. As for the decomposition of semantic and sublinguistic information, it simply employs a validated distillation learning approach for semantic guidance, just to demonstrate that the cutting-edge semantic-sublinguistic decoupling method in current speech codecs can also be synergistically optimized with the proposed speech-background sound decoupling approach. In the future, we will also draw inspiration from subspace de­coupling approaches to further explore more effective semantic-paralinguistic representation methods.

Meanwhile, DeCodec primarily proposes a controllable unified audio representation method and verifies the feasibility of the proposal, while neglecting the representation efficiency of quantizers. In the future, drawing on current research on single-layer quantizer codecs, further optimization of quantizers will also be pursued.

## I    Declaration of AI Use

We used [DeepSeek-V3] to improve the grammatical correctness, fluency, and clarity of the writing. The authors reviewed and approved all the polished content and are responsible for the final content of this manuscript.

