# OpenReview forum: "DeCodec: Rethinking Audio Codecs as Universal Disentangled Representation Learners"
_ICLR.cc/2026/Conference — Submitted to ICLR 2026_

### Official Review · Reviewer_2GYe · 2025-10-27

**Soundness:** 3
**Presentation:** 3
**Contribution:** 2
**Rating:** 4
**Confidence:** 4

**Summary:**

The paper introduces DeCodec, a method to decouple audio representations into orthogonal subspaces for speech and background sound. It further decomposes speech representations into semantic and paralinguistic components, similar to SpeechTokenizer. To achieve this, the authors propose a subspace orthogonal projection module and a representation swap training strategy. The model is trained on 700 hours of speech data mixed with background sounds selected from ESC-50 and DNS-Noise datasets. Experiments demonstrate competitive performance on tasks including speech reconstruction, speech enhancement, and one-shot voice conversion of noisy speech compared to baselines such as EnCodec, HiFi-Codec, DAC, and SpeechTokenizer.

**Strengths:**

1. Decoupling noisy speech into speech and background sound is an interesting and novel idea.
2. The writing is clear and easy to follow.
3. DeCodec inherently enables speech enhancement with performance comparable to specialized models.

**Weaknesses:**

1. The claim of being an "audio codec" is overstated as the experiments only focus on noisy speech data. Typical audio codecs should cover broader audio content, such as pure sounds or music.
2. The study lacks experimental comparisons with relevant works, such as UniCodec and FACodec, which share similar decoupling-based approaches mentioned in the introduction and related work.
3. The experiments are limited to 700 hours of speech data. It is unclear if the method can scale effectively to larger datasets.
4. One-shot voice conversion results are poor (e.g., WER exceeding 50). The qualitative audio demos also exhibit low sound quality. The paper should include comparisons with stronger voice conversion models like CosyVoice2 (e.g., its S3+flow matching method).

Minor Comments:
1. Line 266: Correct "S andnN" to "S and N."
2. Define terms like SRVQ and NRVQ upon first mention.

**Questions:**

Please address the issues mentioned in the Weaknesses section. Resolving these concerns may lead to a higher evaluation.

---

### Official Review · Reviewer_wCbP · 2025-10-27

**Soundness:** 1
**Presentation:** 1
**Contribution:** 1
**Rating:** 2
**Confidence:** 4

**Summary:**

The authors propose, in this paper, the model DeCodec, a disentangled audio codec that separates audio representations into orthogonal subspaces for speech and background sound.  Built on a neural codec framework (DAC model), DeCodec introduces: a subspace orthogonal projection module that factorizes input into decoupled subspaces, and a representation-swap training strategy aligning them with speech and background features. Parallel RVQs independently quantize each subspace, while semantic guidance refines speech disentanglement. Experiments illustrate the performances of DeCodec in terms of reconstruction quality and speech enhancement.

**Strengths:**

The main strengths of the paper are:

-	The concept of using audio codecs to build disentangled representations is very interesting (although not novel)

-	The use of a subspace orthogonal decomposition module to project the primary audio embeddings in two orthogonal subspaces is interesting.

-	The authors provide a comprehensible online demo, illustrating the performance of the proposed model.

**Weaknesses:**

The main weaknesses of the paper are:

-	The authors are largely overstating the results, and in particular regarding the universality of the disentangled representation obtained or in the semantic representation control.

-	The paper is not well positioned with regards to the appropriate State of the art. Some very related works in building disentangled representations using neural audio codecs are not mentioned nor discussed (for example [1] and [2] below). Similarly, the current work is weakly positioned with the SoA in Speech enhancement.

[1] Omran, N. Zeghidour, Z. Borsos, F. de Chaumont Quitry, M. Slaney, and M. Tagliasacchi, “Disentangling speech from surroundings with neural embeddings,” in Proc. IEEE Int. Conf. Acoust., Speech, Signal Process. (ICASSP), 2023.
[2] X. Bie, X. Liu and G. Richard, "Learning Source Disentanglement in Neural Audio Codec," 2025 IEEE International Conference on Acoustics, Speech and Signal Processing (ICASSP), Hyderabad, India, 2025,

-	No indication is given to assess if the differences between approaches are statistically relevant.

-	The justification of the interest of “speech / background sound separation” from the neuroscience results (audio perception exploiting two different auditory cortex regions) is not convincing and appears to be quite disconnected.

-	The clarity and writing of the paper could be largely improved (e.g. figure 1 with almost unreadable texts, many typos or inconsistencies)

**Questions:**

Questions:
-	What are the references of the different speech enhancement models used and compared to (Table 2) ?. Are these models strong baselines ?
-	Typos, some examples: separated, andnN, BRVQ (in figure 2) and NRVQ (in text), Aound, representaions, …

---

### Official Review · Reviewer_Dn8e · 2025-10-29

**Soundness:** 3
**Presentation:** 2
**Contribution:** 2
**Rating:** 2
**Confidence:** 5

**Summary:**

This paper proposes DeCodec, a neural audio codec designed as a universal disentangled representation learner. It introduces three key modules — Subspace Orthogonal Projection (SOP) for separating speech and background sound, Representation Swap Training (RST) for enforcing disentanglement, and Semantic Guidance (SG) using HuBERT features to structure speech representations. The model aims to unify multiple audio tasks (speech enhancement, voice conversion, ASR, TTS) within a single controllable representation space. Experiments on synthetic noisy datasets show strong reconstruction and robustness performance, though evidence for true semantic disentanglement remains mostly empirical rather than theoretical. Overall, DeCodec offers an interesting conceptual reframing and practical architecture, but its theoretical grounding and evaluation depth are limited.

**Strengths:**

* Reinterprets neural audio codecs as universal disentangled representation learners, bridging low-level compression with high-level semantic audio modeling. The bio-inspired analogy to auditory cortex processing is intellectually engaging.

* Simple design: The combination of Subspace Orthogonal Projection (SOP), Representation Swap Training (RST), and Semantic Guidance (SG) forms an elegant, modular framework for factorized representation learning. The approach is lightweight, differentiable, and compatible with existing codecs.

* Unified controllable representations: Demonstrates a single codec that can handle diverse tasks (reconstruction, enhancement, voice conversion, ASR, TTS) through selective subspace manipulation, showing strong versatility and practical controllability.

* Achieves competitive or superior SDR, WER, and DNSMOS scores compared to baselines, particularly in noisy and low-resource conditions, indicating improved robustness and general usability.

* Clear structure: The paper is well-organized, easy to reproduce, and provides intuitive visualizations linking architectural components to perceptual interpretations.

**Weaknesses:**

* Conceptual Validity

The assumption that orthogonality between subspaces guarantees semantic disentanglement is not theoretically grounded.
Orthogonality only enforces statistical decorrelation, not physical or perceptual independence. The learned projections may encode arbitrary frequency or energy partitions rather than truly separating speech and background sound. The analogy with the auditory cortex (A2) is purely metaphorical. Biological “orthogonality” arises from nonlinear, attention-driven and feedback-modulated mechanisms, whereas the proposed SOP relies solely on linear projections and an L2 orthogonality loss.

* Methodological Concerns

Representation Swap Training (RST) lacks theoretical justification. The swap reconstruction objective can be satisfied without genuine factor separation, as the decoder may simply compensate through overparameterization. The observed “disentanglement” might thus emerge from data correlation or reconstruction bias rather than causal structure. The Semantic Guidance (SG) module only supervises the speech branch using HuBERT features but is not jointly optimized with the SOP/RST mechanisms. Therefore, the claimed “hierarchical disentanglement” (semantic vs. paralinguistic) is loosely coupled and not formally guaranteed.

* Evaluation and Empirical Limitations

Experiments are largely performed on synthetic mixtures (LibriTTS + DNS-Noise), without tests on real-world recordings, unseen noise conditions, or cross-domain generalization. Baseline selection is incomplete — comparisons omit recent disentangling codecs such as RepCodec, FunCodec, or SemanticCodec, which pursue similar objectives. The ablation analysis only reports SDR and WER metrics; no interpretability or information-theoretic metrics (e.g., mutual information, alignment scores) are provided to confirm actual semantic separation. Several claims (e.g., controllable feature recombination, semantic robustness) are demonstrated qualitatively through spectrograms, without perceptual or statistical validation. Subjective listening tests (MOS/CMOS/SMOS) are missing; evaluation relies solely on objective or proxy metrics such as DNSMOS and WER, which may not reflect perceptual quality.

* Conceptual Scope and Framing

The “universal front-end” claim is overstated. The multi-task functionality is achieved through post-hoc recombination of representations rather than a truly unified optimization framework. The connection between DeCodec’s representation learning and human auditory cognition is speculative, offering conceptual inspiration but lacking neuroscientific rigor or measurable alignment.

**Questions:**

1. The paper draws inspiration from the auditory cortex (A2) but provides little biological or neuroscientific evidence. Could you clarify whether this analogy is purely conceptual, or if there are empirical findings (e.g., cortical response studies, neural decoding evidence) supporting the proposed “orthogonal subspace” interpretation?

2. Is there any neurophysiological or psychophysical experiment planned (or referenced) to validate the hypothesis that speech and background sound representations in DeCodec correspond to separable cortical processing pathways?

---

### Official Review · Reviewer_JEhp · 2025-11-01

**Soundness:** 2
**Presentation:** 2
**Contribution:** 3
**Rating:** 4
**Confidence:** 3

**Summary:**

The paper introduces DeCodec, which frames the audio codec as a disentangled representation learner. It introduces two mechanisms: Subspace Orthogonal Projection and Representation Swap procedure, to ensure that codecs subspaces correlate to speech and background sound. Experiments show that the codec achieves promising results on downstream tasks such as audio reconstruction, speech enhancement and one-shot voice conversion.

**Strengths:**

Originality: The paper proposed a codec representation via disentanglement to speech and background noise. This disentanglement is achieved via two novel mechanisms: one is enforcing the speech subspace and background subspace as orthogonal, and second via representation swapping which enforces:
(a) Speech Invariance to Background: Speech latent $Zs$ must reconstruct correctly regardless of the noise it is paired with. Hence $Zs$ should only encode speech.
(b) Background invariance to speech:  $Zn$ must reconstruct correctly regardless of speech it is paired with. Hence  $Zn$ should only encode noise.

Quality: The authors conduct thorough experiments for comparing DeCodec against multiple state of the art codec models, such as DAC, SpeechTokenizer for audio reconstruction, and StoRM, SELM for speech enhancement. The authors provide ablations for orthogonal subspace constraint (SOP), representation swapping and semantic guidance. The supplemental material includes visual analysis (cosine similar and singular value comparison), and performance of codec on downstream ASR and TTS tasks to further support the codec's results.

Clarity: The paper is structured well, providing theoretical analysis for factorized audio coding in Section 3, and then providing thorough experimental studies via objective results and visualizations.

Significance: The paper makes a good attempt towards a universal disentangled codec separating the semantic, acoustic and background sounds. It is very relevant to the community working for neural codecs for audio tasks.

**Weaknesses:**

The paper aims towards having audio codec as "universal dis-entagled representation learner", though both the theoretical discussion and experimental results point towards the case in which background noise is added to speech. The results on other audio types like singing/music/sound with more language coverage on evaluation other than English speech should be included to call it universal. Singing and background instruments are co-related and this can break the assumption of orthogonal spaces and representation swapping.

The authors used a well-established way of semantic vs acoustic information separation in neural codecs, which is having Hubert loss on semantic tokens, which limits the novelty to only speech vs background sound separation.

For Table 1, DeCodec bitrate is 8kbps, which is higher than all the baselines. For the experimental results, Table 1 and Table 2, the baseline checkpoints are not re-trained with the new training dataset, which makes a comparison a little unfair.

WER is quite high for Table 3 for Voice Conversion at 50+%, for all baselines, the authors provide an explanation that it's due to different speech segment voicing times, which is not convincing.

**Questions:**

Writing and Notation:
Please increase the font size for the text in the figures. Figure 1 text is really hard to read, especially the ones colored white.

Please consider reducing the amount of new abbreviations like SOP, RST, SG, SRVQ, NRVQ, BRVQ, SDR-O, SDR-B, SDR-S. We could re-name them as Speech-RVQ, Noise-RVQ etc. Background sound vs noise are somewhat used interchangeably in the notation, please keep it consistent to one. The reader should understand tables and figures without referring to the text.

Please have complete headings, instead of ones like "SE", "One-shot VC" in Section 4.

Include the final training loss for the model at end of section 3.6.

Please fix typos such as "representaions", "infered", "aound", "quantinized" etc.

Please use subscript in notation for $Z_s$, $Z_n$, $Z_r$, $Z_c$.

Method and Evaluation:
Please Provide audio samples to convince readers of disentanglement and reconstruction quality. It will be interesting to see if real world noisy data, rather than synthetically created (speech + noise) can be separated well using DeCodec's approach.

It will be interesting to compare the convolution + transformer based architecture choice as seen in Mimi (https://arxiv.org/pdf/2410.00037)t and TAAE (https://arxiv.org/pdf/2411.19842v1) vs a pure convolution architecture based on DAC.

Please explain or include comparison with another common technique for disentanglement ( gradient reversal + supervision) as mentioned in FACodec (https://arxiv.org/pdf/2403.03100), and why SOP + RST is better. The authors claim there is significant information leakage in FACCodec. It would be good to empirically validate that vs DeCodec on real world data.

Please clarify how the Huber semantic guidance is applied.

---

### Official Review · Reviewer_qv8a · 2025-11-01

**Soundness:** 1
**Presentation:** 2
**Contribution:** 2
**Rating:** 2
**Confidence:** 4

**Summary:**

This paper proposes a neural audio codec, DeCodec, designed to disentangle speech and background sounds into separate, orthogonal representations. To achieve this, the authors introduce a subspace orthogonal projection (SOP) module and a representation swap training (RST) strategy.

**Strengths:**

The goal of disentangling audio components within a codec is interesting and, if successful, could benefit downstream tasks like speech enhancement, noise-robust voice clone.

**Weaknesses:**

1. Fundamental flaw in premise: The core assumption that speech and background must be orthogonal is ill-posed for a reconstruction task. It does not convince me that there is no overlapping in the acoustic phonomenons of speech and background sound. Enforing the codec to encode them into orthogonoal representations could incur significant distortion in the decoupled outputs, as the decoder is deprived of shared acoustic information.
2. Poor reconstruction quality on individual representation: The audio samples on the demo page substantiate the theoretical concern. Both the extracted speech and background sound in speech enhancemnet scenario are noticeably distorted, indicating that the decoder still relies on the combination of the separate representations to generate high-fidelity audio.
3. Inefficient representation compression: The decouple of the audio signal in the proposed neural audio codec does not help the compression of encoded representation, but doubles the bitrate to 8kbps instead! Considering that the separate representation is not properly decoupled, the added overhead is not dispensible when applied to various downstream tasks, especially for generation.
4. The semantic guidance mechanism for the speech encoder has been established in prior work and does not constitute a novel contribution.
5. There major presentation issues in the paper:
* In all figures and tables, the VQ module of DeCodec contains a SRVQ and a BRVQ. While in the main text, the BRVQ is never mentioned, but replaced with a NRVQ without any explanation.
* The font size of Figure 1 is too tiny to read easily.
* The Figure 2 is missing a lot of captions and legends to enable a basic understanding of the proposed workflow.

**Questions:**

1. It is claimed that the paper proposes a novel neural codec that learns to decouple audio representations into “orthogonal subspaces” dedicated to speech and background sound. Although the authors give some justifications about "orthogonal" on Section 3.4 when introducing SOP module, the experimental results can not give full support that the speech and background sound features are really "orthogonally" disentangled.
2. What is the theoretical or empirical justification for the requirement of orthogonal representations? Given the shared acoustic properties of speech and noise, how can high-fidelity reconstruction be achieved from a single, artificially constrained representation?
3. How does this method aid compression, given that it doubles the bitrate? What is the computational overhead (e.g., latency, memory) when integrating DeCodec with a downstream LLM for a task like TTS?
4. In downstream TTS evalution results, why is the comparison with DAC missing (which is compared in ASR evalution) ? There is only the weak baseline SpeechTokenizer which shows inferior reconstruction fidelity to DAC.
5. The main text refers to an "NRVQ," while the figures label a "BRVQ." Are these the same component? If not, please clarify the difference.

---

### Meta-Review · Area_Chair_SDek · 2026-01-06

**Summary:**

This paper proposes DeCodec, a neural audio codec designed to disentangle audio representations. It involves two mechanisms, Subspace Orthogonal Projection and the Representation Swap procedure, to ensure that codec subspaces correlate with speech and background sound.

Reviewers found the paper’s high-level motivation of rethinking audio codecs as disentangled representation learners to be interesting and potentially impactful (qv8a, wCbP, 2GYe). The idea of decomposing audio into orthogonal subspaces for background sounds and multiple speech attributes was generally appreciated, and the paper explores a broad set of downstream tasks.

However, the reviews raised substantial concerns. In particular, reviewers found that comparisons to relevant and recent baselines were incomplete, and that the evaluation does not sufficiently demonstrate the advantages of the proposed design over existing neural codec and disentanglement approaches (qv8a, wCbP, 2GYe). Meanwhile, reviewers found that the claim of DeCodec as a universal audio codec is overclaimed and lacks supporting experiments (Dn8e, JEhp, wCbP, 2GYe). In addition, reviewers expressed concerns about flawed motivations and assumptions (qv8a, Dn8e, wCbP), which further weaken the paper’s claims.

Taken together, while the research direction is promising, the current submission does not provide enough empirical or conceptual grounding to support acceptance at this time.

**Reviewer Concerns:**

The authors did not provide a rebuttal. The major concerns regarding baseline comparisons, overstatement, and flawed motivations and assumptions remain unresolved.

**Reviewer Scores:**

Without a rebuttal, the reviewers would likely maintain their current scores. Most scores would likely remain at 2.

---

### Decision · Program_Chairs · 2026-01-26

Reject